# Comparison of Proton and Gamma Irradiation on Single-Photon Avalanche Diodes

Mingzhu Xun [1,2,3], Yudong Li [1,2,*] and Mingyu Liu [1,2,3]

1   State Key Laboratory of Functional Materials and Devices for Special Environmental Conditions, Xinjiang Technical Institute of Physics and Chemistry, Chinese Academy of Sciences, Urumqi 830011, China; xunmz@ms.xjb.ac.cn (M.X.); liumingyu21@mails.ucas.ac.cn (M.L.)
2   Xinjiang Key Laboratory of Extreme Environment Electronics, Urumqi 830011, China
3   University of Chinese Academy of Sciences, Beijing 100049, China
*   Correspondence: lydong@ms.xjb.ac.cn

**Abstract:** In this paper, the effects of proton and gamma irradiation on reach-through single-photon avalanche diodes (SPADs) are investigated. The I–V characteristics, gain and spectral response of SPAD devices under proton and gamma irradiation were measured at different proton energies and irradiation bias conditions. Comparison experiments of proton and gamma irradiation were performed in the radiation environment of geosynchronous transfer orbit (GTO) with two different radiation shielding designs at the same total ionizing dose (TID). The results show that after 30 MeV and 60 MeV proton irradiation, the leakage current and gain increase, while the spectral response decreases slightly. The leakage current degradation is more severe under the "ON"-bias condition compared to the "OFF"-bias condition, and it is more sensitive to the displacement radiation damage caused by protons compared to gamma rays under the same TID. Further analysis reveals that the non-elastic and elastic cross-section of protons in silicon is $1.05 \times 10^5$ times greater than that of gamma rays. This results in SPAD devices being more sensitive to displacement radiation damage than ionizing radiation damage. Under the designed shielding conditions, the leakage current, gain and spectral response parameters of SPADs do not show significant performance degradation in the orbit.

**Keywords:** SPADs; proton irradiation; gamma irradiation; I–V characteristics

## 1. Introduction

A SPAD is a photodetector capable of detecting and counting single photons. By reversing the bias of the PN junction and operating above the breakdown voltage, photo-electrons generated by the photoelectric effect experience continuous multiplication in a strong electric field when a single photon impacts the multiplication region of the SPAD, resulting in an amplified photocurrent [1,2]. The photocurrent can be converted into a voltage signal for preamplifying or counting by an active or passive external quenching circuit. A SPAD has the advantages of high sensitivity, high gain, high temporal resolution, magnetic resistance and low noise for single-photon detection, and in recent years, it has been widely used in a variety of weak optical signal detection fields, including star-ground quantum communication, quantum key distribution, quantum computation, and high-energy particle detection in space, atmospheric pollution monitoring, astronomical observation, photoelectric imaging and other space missions [3–5].

In 1963, R.H. Haitz et al. clarified the working principle of SPADs, explained the physical phenomena generated by photon pulses, dark current pulses and residual pulses, and proposed the theoretical model of photon pulses [6,7]. They also designed a SPAD device model. In 1999, McIntyre et al. established a comprehensive theoretical model for the impact ionization process of SPADs [8,9]. After 60 years of rapid development, the materials, structures and circuits of SPADs have been optimized, and the type and performance of

SPADs have been significantly improved, accelerating the large-scale application of single-photon detectors in fields such as quantum communication, light detection and ranging (LIDAR), and medical detection [5,10–13]. The key to the performance improvement is the availability of high-quality materials with higher purity and fewer defects, as well as novel SPAD structures. Structurally, the types of detectors that have been developed include Si-SPADs, InGaAs detectors, Ge detectors, SiC detectors, superconducting nanowire single-photonic detectors (SNSPDs) and quantum-dot optically gated field-effect transistors (QDOGFETs) [14–19]. Different SPAD materials have different sensitivities to wavelengths. For example, GaN-based SPADs are sensitive to ultraviolet (UV) light and are typically used to measure weak UV light. Due to the bandwidth of silicon, Si-SPADs have a limited operating range and can only be used in the visible and near-infrared wavelengths. For infrared wavelengths above 1100 nm, materials with a bandgap of less than 1.1 eV should be used, typically In-GaAs and InP materials [20]. Structural innovations include back-illuminated SPADs, vertical double-junction SPADs, substrate-isolated SPADs, 3D-stacked SPADs, non-contact ring guard SPADs and high-detection-efficiency SPADs [21,22]. These novel materials and structures further improve SPADs' detection efficiency and reduce dark counts.

With the large-scale commercialization and application of low-cost devices in the space field, SPADs are inevitably exposed to proton and electron radiation in the space radiation environment, leading to radiation damage to the SPAD. Recent studies have shown that the radiation effect on Si-based devices mainly causes TID and displacement damage dose (DDD) [23,24]. The energy loss of incident particle ionization leads to the generation of electron hole pairs in irradiated materials, while non-ionization energy loss leads to phonon generation and lattice atomic displacement, resulting in ionization and displacement damage [25,26]. These damages can form movable charged defects, interface states defects and Frenkel defects in silicon-based devices, which can act as carrier traps and recombination centers [27,28]. This damage can increase the leakage current, dark count rate (DCR) and equivalent noise of the SPAD, and potentially affect the performance and lifetime of the SPAD.

To investigate the radiation effects on SPADs, we conducted a detailed analysis of the radiation damage inflicted by protons and gamma rays on SPAD devices. The radiation environments of specific orbit altitude and radiation shielding designs were considered. Furthermore, the differences and similarities in the degradation of SPAD parameters were compared. In this experiment, back-illuminated SPADs were selected. The changes in dark current, gain and spectral response of the SPAD after proton and gamma irradiation were analyzed in detail. The degradation of the SPAD parameters under equivalent TID for proton and gamma radiation were compared. From the perspectives of the number of displacement damage defects and collision cross-sections, the SPAD degradation mechanisms of proton and gamma radiation were explained.

## 2. Experimental Design

The cross-section of the back-illuminated SPADs used in this experiment is shown in Figure 1 [29]. The SPADs have a thick depletion layer structure, and the multiplication region is composed of a PN junction formed by the contact between phosphorus-doped n+-type and boron-doped p-type semiconductors, as well as a lightly doped phosphorus-doped n-type guard ring around the edges of the PN junction [30]. The guard ring structure prevents edge breakdown, reduces surface tunneling, and improves photon detection efficiency. The distance from the wafer surface to the depletion region of the PN junction is reduced to 30–40 μm in thickness through a wafer back etching process, and a low-doped, low-resistance light absorption region is formed using a boron diffusion process. When photons impact in the absorption layer and generate photoelectrons, the photoelectrons drift into the multiplication region under the reverse electric field. The strong electric field ($E > 1 \times 10^5$ V/cm) in the multiplication region accelerates the photoelectrons to obtain sufficient kinetic energy, and the photoelectrons transfer energy to the electrons of new

lattice atoms, ionizing them into secondary electrons through the Coulomb interaction. The photoelectrons and secondary electrons continuously multiply in the multiplication region, forming a sustainable multiplication current. The multiplication current can be converted into a voltage signal for subsequent signal amplification and pulse counting by an external active or passive quenching circuit. The light-absorbing region of this SPAD structure is relatively thick, usually 30–40 μm. A high breakdown voltage is required to achieve the avalanche breakdown condition, usually above 200 V [31]. Due to the use of epitaxial manufacturing processes, the lattice defects, point defects and impurity concentrations in the SAPD are lower. But the remaining defects in the multiplication region cause Shockley–Read–Hall (SRH) recombination and trap-assisted tunneling, resulting in the random release of carriers and an increase in the DCR [29]. Therefore, the SPAD needs to operate at a low temperature of −20 °C to reduce the DCR. The breakdown voltage of the SPAD selected for this experiment is about 225 V, and the reverse leakage current is about 7 nA. A reach-through structural SPAD was selected for gamma and proton radiation effect research due to its simple structure, mature manufacturing process and complete theoretical model for analyzing I–V and spectrum characteristics [31,32].

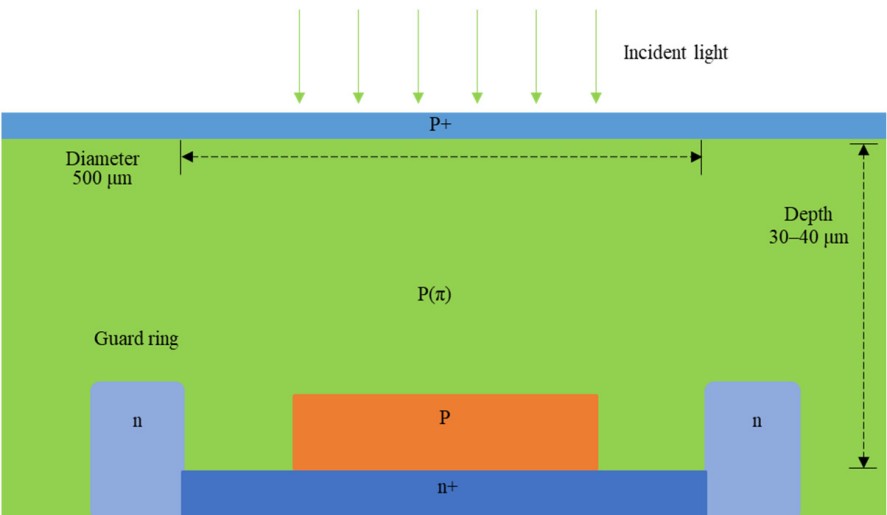

**Figure 1.** Cross-section of the reach-through structure of the SPAD.

In order to analyze the effect of TID and DDD on the SPAD in a spatial radiation environment, this experiment was designed based on the radiation environment at the GTO, with an altitude of 35,786 km, corresponding to a mission duration of four years in the orbit. Two shielding materials were designed: a 3.705 mm (1.0 g/cm$^2$) aluminum shield material, and 8.234 mm Al with a 0.467 mm Ta shield material (3.0 g/cm$^2$ Ta to Al with a mass ratio of 35%). After the simulation of space radiation particle transport in the shielding material by Geant4, which is a toolkit for the simulation of particles through matter invented by the European Organization for Nuclear Research (CERN) [33–35], it was calculated that the energies corresponding to the differential energy spectrum peaks of the protons (including primary and secondary protons) generated by solar protons after passing through the two shielding materials are approximately 30 MeV and 60 MeV, as shown in Figure 2. The TID and DDD were also calculated as shown in Table 1. In addition to particles produced by solar protons, protons and electrons from the Van Allen radiation belts and solar flare outbursts also contribute to TID and DDD. The contribution of radiation belts and solar flare outbursts to TID and DDD was taken into account in the design of this experiment. Therefore, the TID and DDD corresponding to the total proton flux used in the experiment exceed the shielding design values shown in Table 1, as detailed in Table 2.

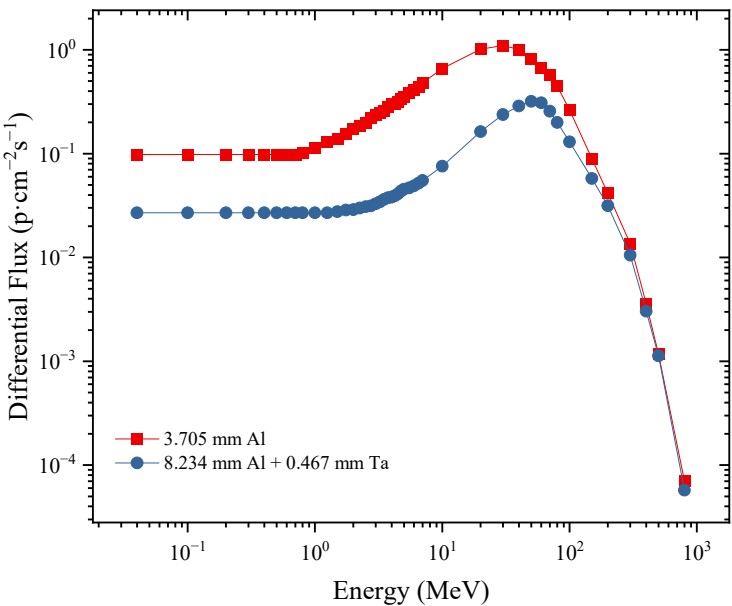

**Figure 2.** Proton differential flux of two designed shielding materials.

**Table 1.** TID and DDD of two designed shielding materials at the GTO for 4 years.

| Designed Shielding Material | TID (krad) | DDD (MeV/g) |
|---|---|---|
| 3.705 mm Al | 1.35 | $2.84 \times 10^7$ |
| 8.234 mm Al + 0.467 mm Ta | 0.346 | $7.02 \times 10^6$ |

**Table 2.** Proton fluence, LET, NIEL, TID and DDD of 30 MeV and 60 MeV protons used in experiment.

| Proton Energy (MeV) | LET (MeV/(g/cm$^2$)) | NIEL (MeV/(g/cm$^2$)) | Fluence (p/cm$^2$) | TID (krad) | DDD (MeV/g) |
|---|---|---|---|---|---|
| 30 | 14.76 | $5.63 \times 10^{-3}$ | $1.00 \times 10^9$ | 0.235 | $5.63 \times 10^6$ |
| | | | $5.00 \times 10^9$ | 1.175 | $2.82 \times 10^7$ |
| | | | $1.00 \times 10^{10}$ | 2.35 | $5.63 \times 10^7$ |
| | | | $5.00 \times 10^{10}$ | 11.75 | $2.82 \times 10^8$ |
| | | | $1.00 \times 10^{11}$ | 23.5 | $5.63 \times 10^8$ |
| 60 | 8.6 | $2.89 \times 10^{-3}$ | $1.00 \times 10^9$ | 0.138 | $2.89 \times 10^6$ |
| | | | $5.00 \times 10^9$ | 0.69 | $1.45 \times 10^7$ |
| | | | $1.00 \times 10^{10}$ | 1.38 | $2.89 \times 10^7$ |
| | | | $5.00 \times 10^{10}$ | 6.9 | $1.45 \times 10^8$ |
| | | | $1.00 \times 10^{11}$ | 13.8 | $2.89 \times 10^8$ |

The proton testing was conducted at the Proton Radiation Effect simulation test Facility (PREF) of the Xinjiang Institute of Physics and Chemistry, Chinese Academy of Sciences. The proton energy was 30 MeV and 60 MeV and the beam area was 2 cm × 2 cm, with a proton fluence of $4.79 \times 10^8$ p/cm$^2$. The proton uniformity was within ±5%. The proton fluence, line energy transport (LET), non-ionizing energy loss (NIEL), TID and DDD of the experiment are shown in Table 2. The LET data were obtained from the National Institute of Standards and Technology (NIST)'s stopping power and range tables for protons (PSTAR) program, and the NIEL data were obtained from reference [36]. The gamma irradiation was conducted at the Co-60 gamma source of the Xinjiang Institute of Physics and Chemistry, Chinese Academy of Sciences. The dose rate was 50 rad(Si)/s, with a cumulative dose of 70 krad(Si).

Three bias conditions were applied during the irradiation: "ON$_{M=100}$"-bias condition, "ON$_{M=10}$"-bias condition and "OFF"-bias condition. The "ON$_{M=100}$" bias applies an op-

erating voltage of 215 V, and at this voltage, the gain of the SPAD is 100 (M = 100). The "$ON_{M=10}$" bias applies an operating voltage of 180 V and the gain of the SPAD is 10 (M = 10). The "OFF" bias applies an operating voltage of 0 V and all pins are shorted together and grounded. All parameter measurements were completed within 2 h after irradiation.

The schematic diagram of the electrical parameter testing system used is shown in Figure 3. The halogen lamps provide a stable light source with a wavelength of 350–1100 nm. After passing through the filter, a single wavelength light is generated in the spectrograph. The integrating sphere performs multiple diffuse reflections on the incident light to improve the uniformity of the light incident on the test sample, ensuring photostability during sample testing. In terms of parameter measurements, the Keysight semiconductor parameter analyzer was used to measure the I–V characteristics of the device before and after irradiation, and an oscilloscope was used to measure and observe the SPAD's pulse output waveform. The external test circuit consisted of a passive quenching circuit with a 50 kΩ resistor. An external counter was used to accurately measure the dark count.

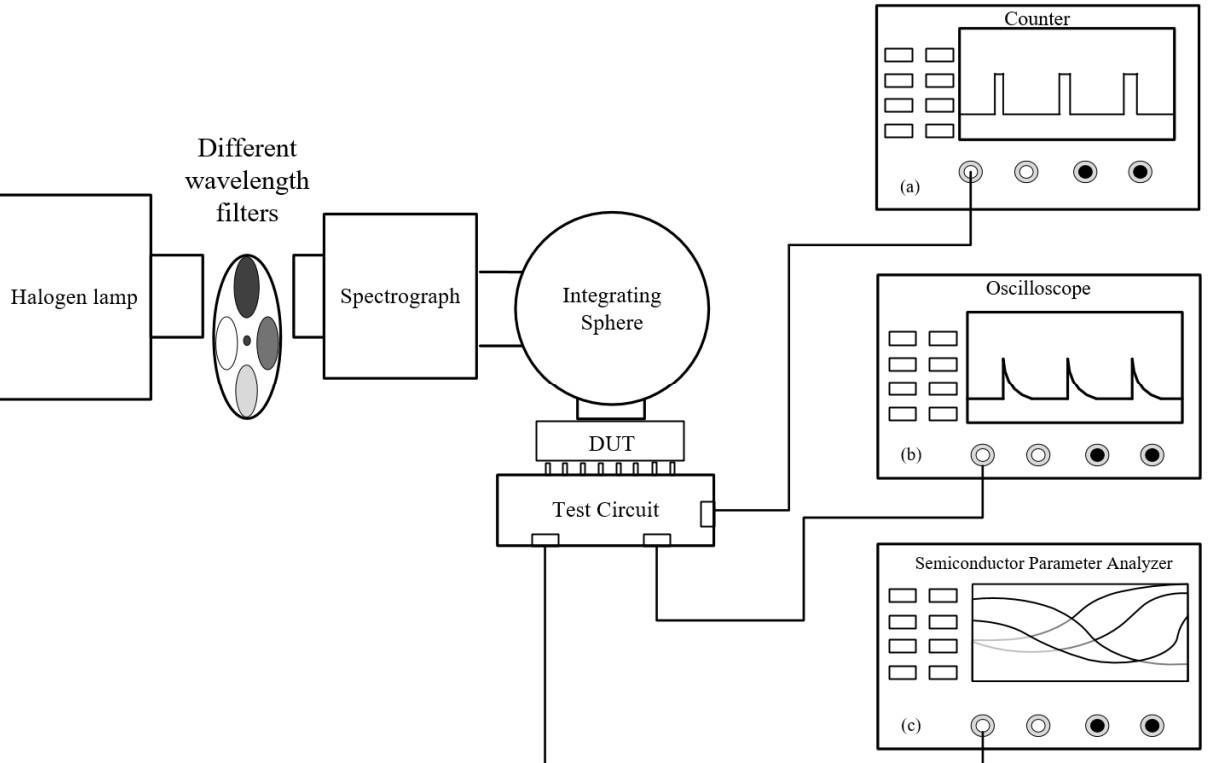

**Figure 3.** The SPAD parameter testing system with (**a**) a counter for the dark count rate (DCR) measurement, (**b**) an oscilloscope for pause shape observations and (**c**) a semiconductor parameter analyzer for I–V characteristic measurements [37].

## 3. Results

### 3.1. I–V Characteristics

The I–V characteristics of the SPAD before and after gamma radiation are shown in Figure 4. Figure 4a shows that the leakage current increases with increasing radiation dose under the "OFF"-bias condition. This is measured in a dark environment. At a reverse voltage of 200 V, the leakage current increases from 5.79 nA before irradiation to 6.8 nA after accumulating 70 krad(Si) TID, an increase of 1.01 nA. When the accumulated TID reaches 2.35 krad(Si), which corresponds to a proton fluence of $1 \times 10^{10}$ p/cm$^2$, the leakage current at a reverse voltage of 200 V increases to 6.173 nA, an increase of 0.383 nA. This indicates that the leakage current does not increase significantly under "OFF"-bias conditions. However, under the "$ON_{M=100}$"-bias condition, as shown in Figure 4b, the leakage current increases from 6.39 nA to 20.94 nA after accumulating a TID of 30 krad(Si)

at a reverse voltage of 200 V. When the accumulated TID reaches 70 krad(Si), the leakage current increases to 12.26 μA. This implies that the SPAD is severely damaged. The increase in leakage current of the "ON$_{M=100}$"-bias irradiated SPAD is more significant compared to the "OFF"-bias condition at the same TID.

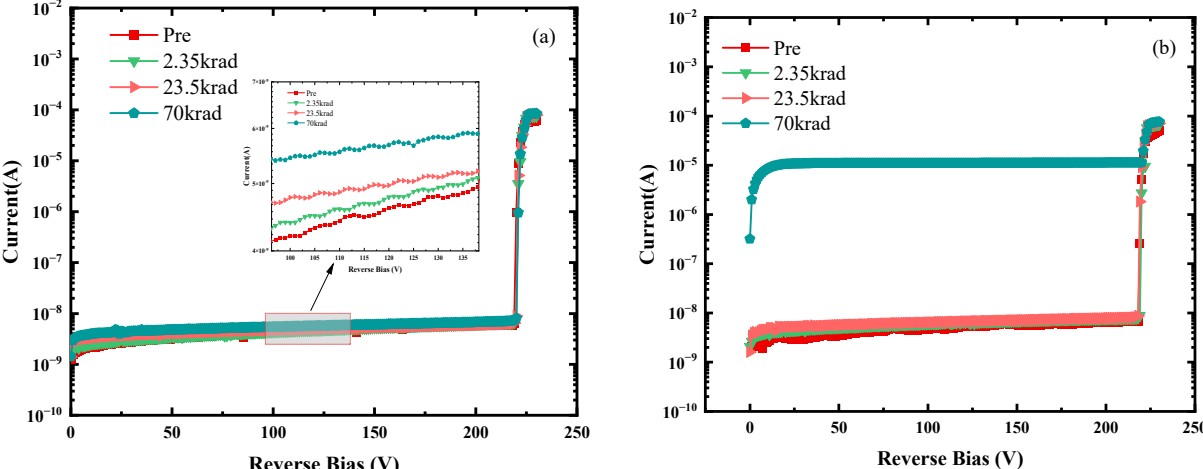

**Figure 4.** I–V characteristics of SPAD after γ irradiation under the (**a**) "OFF"-bias condition and (**b**) "ON$_{M=100}$"-bias condition.

The I–V characteristics of the SPAD before and after 30 MeV proton irradiation are shown in Figure 5. As shown in Figure 5a, the leakage current increases slightly with the increase in proton irradiation fluence under the "OFF"-bias condition. At a reverse voltage of 80 V, the leakage current increases from 3.49 nA to 4.66 nA at a fluence of $5 \times 10^{10}$ p/cm$^2$. However, when the reverse voltage reaches 117 V, the leakage current exhibits a "step" increase, similar to the "punch-through" phenomenon between the multiplication region and the absorption region [38]. At a reverse voltage of 200 V, the reverse leakage current increases from 5.42 nA before irradiation to 24.7 nA after irradiation. It can be seen that as the reverse voltage increases, the leakage current also increases, but the breakdown voltage of the SAPD device does not change significantly before and after irradiation. The results of proton irradiation under the "ON$_{M=100}$"-bias condition are shown in Figure 5b. The leakage current also gradually increases with the accumulation of fluence. However, for the $1 \times 10^{10}$ p/cm$^2$ proton fluence required by the on-orbit shielding design, the leakage current of the SPAD device increases from 5.68 nA before irradiation to 14.3 nA after irradiation at an operating voltage of 200 V. But when the proton fluence accumulates to $5 \times 10^{10}$ p/cm$^2$, the leakage current suddenly increases to 1.82 μA, which means there is serious damage. This indicates that under proton irradiation, with an "ON$_{M=100}$"-bias condition, the damage to the SPAD is more severe compared to that caused by the "OFF"-bias condition.

To further analyze the effect of bias conditions on the reverse leakage current under different proton irradiation energies, we conducted proton irradiation with 30 MeV and 60 MeV protons at a cumulative fluence of $1 \times 10^{11}$ p/cm$^2$. Three bias conditions were applied during the irradiation: the "ON$_{M=100}$"-bias condition, "ON$_{M=100}$"-bias condition and "OFF"-bias condition. The measured I–V characteristics are shown in Figure 6. From the test results, it can be observed that under the same proton energy, the leakage current increases with the increase in bias gain, indicating that the number of defects produced by proton incidence in the SPAD increase with the increase in the electric field in the multiplication region. Under the "OFF"-bias condition, both proton energies exhibit a "step" increase near the voltage of 117 V, which does not occur under the conditions of gain of 10 and 100. The reason may be that under the condition of no electric field, the charged defects generated during irradiation cannot migrate due to the absence of an electric field, resulting in local accumulation. This creates a local built-in electric field

opposite to the direction of the electric field during device operation. This built-in electric field hinders the transport of charge carriers, reducing the leakage current. As the voltage increases to a certain level, the charges trapped by the defects are released under the strong electric field, reducing or even eliminating the built-in electric field, leading to a sudden increase in the leakage current. Under both 30 MeV and 60 MeV irradiation energies, the reverse leakage current under the three bias conditions was higher for 30 MeV proton irradiation than for 60 MeV proton irradiation. This is due to the higher LET and NIEL values for 30 MeV proton irradiation compared to 60 MeV, as shown in Table 2. The higher defect production by 30 MeV protons results in more severe radiation damage to the SPAD. Further analysis reveals that under the condition of gain of 100, the effect of proton displacement damage dose (DDD) on the leakage current of the SPAD is greater than the effect of total ionizing dose (TID). The TID for 30 MeV protons at a fluence of $1 \times 10^{11}$ p/cm$^2$ is 23.5 krad(Si). However, as shown in Figure 4b, the leakage current after gamma irradiation at this dose is less than 10 nA. In contrast, when proton irradiation is carried out at a fluence of $1 \times 10^{11}$ p/cm$^2$, which corresponds to a TID of 23.5 krad(Si) and a DDD of $5.63 \times 10^8$ MeV/g, the leakage current increases by a factor of 1000 to more than 10 μA. This demonstrates that compared to total ionizing damage, the leakage current of the SPAD is more sensitive to displacement damage.

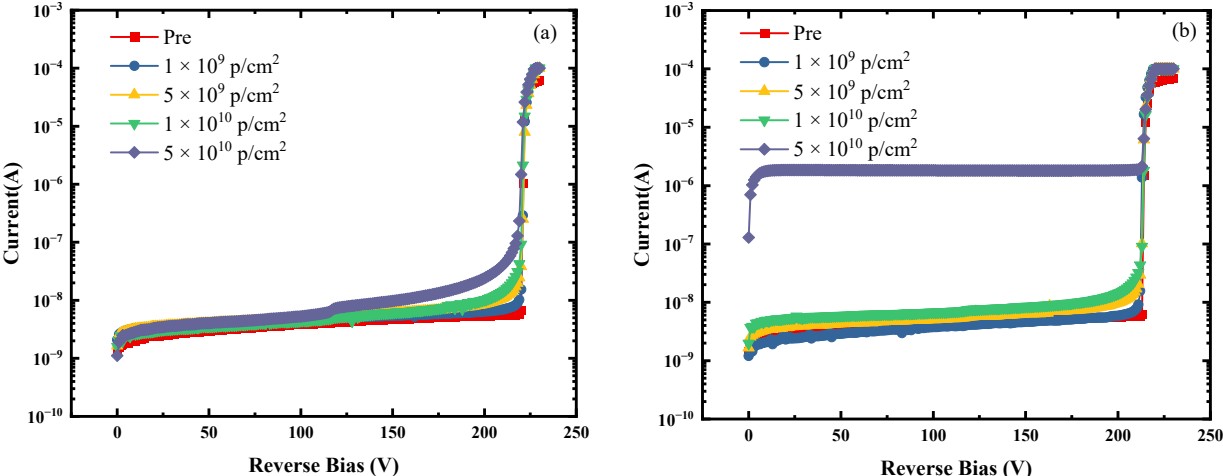

**Figure 5.** I–V characteristics of SPAD after 30 MeV proton irradiation under the (**a**) "OFF"-bias condition and (**b**) "ON$_{M=100}$"-bias condition.

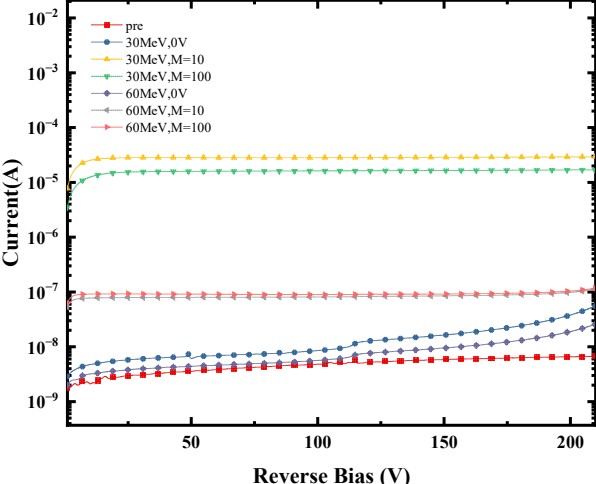

**Figure 6.** I–V characteristics of SPAD after proton irradiation under different bias conditions of 30 MeV and 60 MeV.

### 3.2. Multiplication

The variation in gain versus reverse voltage after gamma and proton irradiation is shown in Figure 7. Here, the gain is defined as the ratio of the photocurrent with bias voltage to the photocurrent without bias voltage. Figure 7a represents the gain curve after gamma irradiation. It can be observed that under the "OFF"-bias condition, as the accumulated TID increases, the SPAD gain remains almost unchanged. This indicates that at gamma irradiation doses up to 70 krad(Si), the photocurrent generation capability is not significantly improved compared with that before irradiation. The defects produced by gamma irradiation do not affect the carrier drift process in the absorption region or the avalanche process in the multiplication region of the SPAD device. Therefore, the gain remains almost unchanged after gamma irradiation.

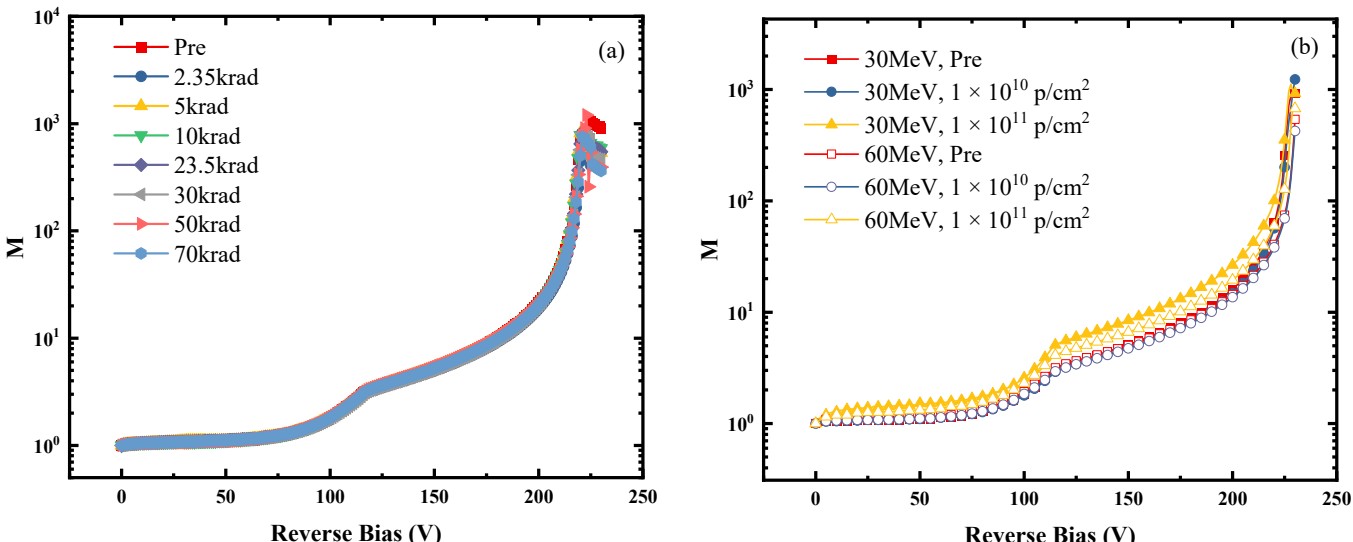

**Figure 7.** Multiplication of SPAD after (**a**) $\gamma$ irradiation and (**b**) 30 MeV and 60 MeV proton irradiation.

Figure 7b shows the gain variation at different reverse voltages after 30 MeV and 60 MeV proton irradiation. From the figure, it can be seen that at a fluence of $1 \times 10^{10}$ p/cm$^2$, the gain of the SPAD remains almost constant at both energies. However, when the accumulated fluence reaches $1 \times 10^{11}$ p/cm$^2$, the gain starts to increase, and the gain after 30 MeV proton irradiation is more significant than that after 60 MeV proton irradiation, which indicates that 30 MeV protons cause more serious radiation damage to the SPAD. By observing the change in photocurrent and dark current before and after irradiation, it was found that both currents increased after irradiation. However, the dark current $I_{dark}$ and the $I_{unit,dark}$ both increased significantly compared with those before irradiation. The increase in the $I_{unit,dark}$ is more significant, resulting in a decrease in the photocurrents without bias voltage after irradiation, and a consequent increase in the gain. The gain ($M$) is calculated as follows:

$$M = \frac{I_{photo} - I_{dark}}{I_{unit,photo} - I_{unit,dark}}$$

In the formula, $I_{photo}$ is the photocurrent measured in the presence of light and with bias voltage, $I_{dark}$ is the leakage current measured in the absence of light but with bias voltage, $I_{unit,photo}$ is the photocurrent measured in the presence of light but without bias voltage, and $I_{unit,dark}$ is the leakage current measured in the absence of light and without bias voltage.

### 3.3. Spectral Response

Figure 8a displays the spectral response after gamma irradiation under the "ON$_{M=100}$"-bias condition. It is worth noting that the spectral response of the SPAD with an incident

wavelength of less than 700 nm remains relatively stable during irradiation with a total cumulative dose of 30 krad(Si). However, in the wavelength range of 700–900 nm, the spectral response fluctuates, showing an increase under a TID of 30 krad(Si) followed by a decrease under a TID of 50 krad(Si). This behavior may be attributed to variations in carrier release and multiplication caused by mobile charged defects generated by ionizing radiation, inherent device defects and displacement damage defects resulting from a high TID.

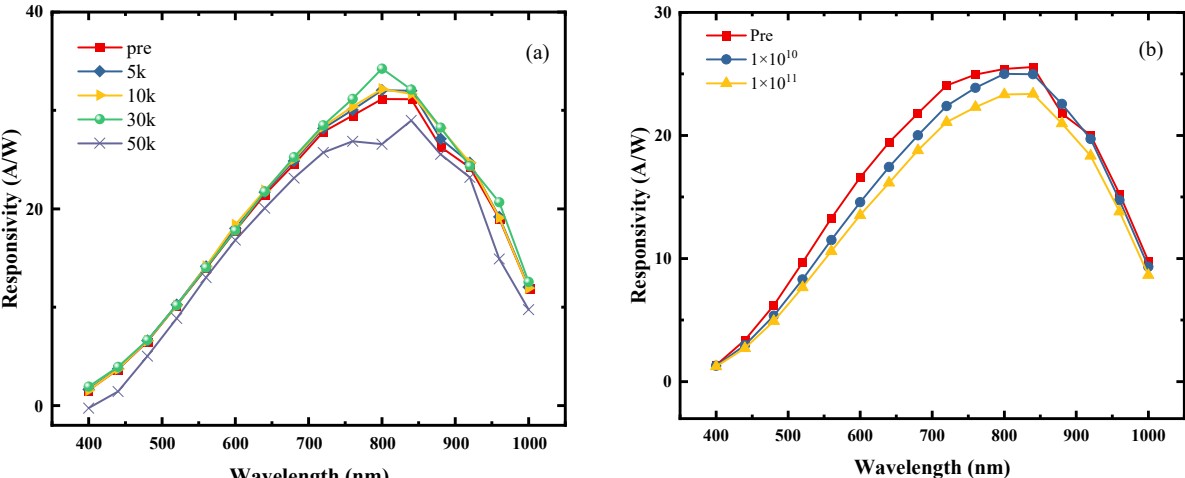

**Figure 8.** Spectral response of SPAD after (**a**) $\gamma$ irradiation and (**b**) 60 MeV proton irradiation under "OFF"-bias condition.

Figure 8b shows the spectral response after 60 MeV proton irradiation under the "OFF"-bias condition. When the proton irradiation fluence reaches $1 \times 10^{10}$ p/cm$^2$, the spectral response curve decreases significantly at wavelengths of 400–850 nm, and when the proton fluence reaches $1 \times 10^{11}$ p/cm$^2$, the spectral response decreases at wavelengths in the range of 400–1000 nm. The photocurrent of the device is reduced due to the effect of displacement damage defects formed by proton irradiation in the photoelectron multiplication process, leading to a decrease in spectral response. Reference [39] reported spectral response degradation after neutron irradiation of avalanche photodiodes (APDs), and the neutron energy was 1 MeV, the neutron flux was $2 \times 10^{13}$ n/cm$^2$ and the cumulative DDD was $4.07 \times 10^{10}$ MeV/g. The peak spectral response at a wavelength of 600 nm decreased from 23.5 A/W to 9.5 A/W after irradiation, with a gain of 40. This indicates that displacement damage produced by protons and neutrons would also cause spectral response degradation.

## 4. Discussion

Based on the previous experimental results, it can be observed that compared to the ionization radiation damage caused by gamma irradiation, the displacement radiation damage inflicted by protons on SPADs leads to more pronounced radiation-induced degradation in the leakage current, gain and spectral response. Previous research has shown that gamma radiation in silicon-based devices primarily produces ionization radiation damage through Compton scattering and displacement damage due to collisions between secondary high-energy electrons and silicon [40]. On the other hand, protons cause ionization radiation damage through ionization and excitation processes, and displacement damage through elastic and inelastic collisions (nuclear reactions) between protons and silicon atoms. These ionization and displacement radiation damage processes generate defects such as vacancy ($V_{Si}$), substitutional phosphorus ($P_{Si}$), interstitial oxygen ($O_i$), double vacancy ($V_{Si}$-$V_{Si}$), A-center ($V_{Si}$-$O_i$) and E-center ($P_{Si}$-$V_{Si}$) [27,28,41–43]. Displacement radiation damage-induced vacancy defects can form deep-level defects within the SPAD's multiplication region, capturing and releasing carriers as generation–

recombination centers. They randomly emit electrons to the multiplication area, causing increased leakage currents.

Due to the more severe parameter degradation observed in SPADs after proton irradiation, the Stopping and Range of Ions in Matter Software 2013 (SRIM-2013) is used to analyze the displacement radiation damage caused by 30 MeV and 60 MeV protons [44]. This software performs Monte Carlo simulations of ion transportation; through detailed calculations, the number of vacancy defects generated per particle, the energy deposition, the particle range and other relevant information can be obtained. The simulation results for 30 MeV and 60 MeV protons are presented in Table 3. It can be seen that 30 MeV protons produce a displacement damage vacancy fluence of 24 n/(ion·mm) in silicon, while 60 MeV protons produce only 10.4 n/(ion·mm). This explains why the parameter degradation after 30 MeV proton irradiation is more severe than that caused by 60 MeV proton irradiation. Additionally, Figure 9 shows the depth dose distribution curves, indicating that although 60 MeV protons have a deeper penetration depth, within the effective thickness of the device (~200 μm), 30 MeV protons exhibit a higher stopping power at the SAPD surface, depositing more energy and generating a higher number of vacancy defects. This leads to more severe parameter degradation under the same proton irradiation conditions.

**Table 3.** Comparison of the SPAD breakdown voltage.

| Proton Energy (MeV) | Total Vacancies (n/(ion·mm)) | Total Stopping Power (MeV/g) | Projected Range (mm) |
|---|---|---|---|
| 30 | 24 | $3.441 \times 10$ | 4.91 |
| 60 | 10.4 | $2.004 \times 10$ | 16.85 |

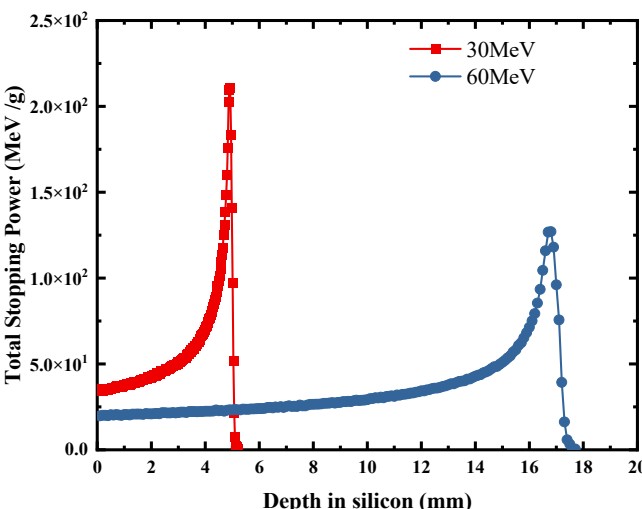

**Figure 9.** Depth dose of 30 MeV and 60 MeV protons in silicon.

The internal structure of the SPAD devices discussed does not have a large Si-SiO$_2$ interface. Therefore, migration of ionizing radiation damage defects to the interface due to gamma irradiation does not result in the formation of a large number of interfacial states [45,46]. The parameter changes observed after gamma irradiation are mainly attributed to displacement damage defects formed by energetic secondary electrons generated by the interaction of gamma rays with lattice atoms [25,47].

In fundamental studies of ray–matter interactions, the reaction cross-section is used to characterize the probability of interactions between rays and matter. Physically, it represents the probability of a nuclear reaction occurring when a single particle impinges on a unit area containing only one target nucleus. The unit of the cross-section is $1b = 10^{-24}$ cm$^2$. By referring to the TENLDL-2023 basic nuclear database published by the Paul Scherrer Institute (PSI) and comparing gamma ray and proton interactions and nuclear reaction

processes on silicon materials, we obtain the data presented in Table 4. It can be observed that gamma rays primarily suffer from elastic collisions with silicon atomic nuclei when entering silicon, without non-elastic scattering. This means that the displacement damage is caused by elastic collisions between gamma photons and atomic nuclei, without involving nuclear reactions. The Compton effect between photons and outer shell electrons is not included in this elastic scattering cross-section.

**Table 4.** Elastic and non-elastic cross-section of gamma rays and protons in silicon.

| Incident Particle | Particle Energy (MeV) | Non-Elastic Cross-Section(mb) | Elastic Cross-Section (mb) |
|---|---|---|---|
| gamma | 1 | 0 | $2.801 \times 10^{-3}$ |
| | 1.25 ($^{60}$Co) | 0 | $5.522 \times 10^{-3}$ |
| | 2 | 0 | $1.368 \times 10^{-2}$ |
| proton | 30 | $7.428 \times 10^2$ | $4.835 \times 10^{-3}$ |
| | 60 | $5.778 \times 10^2$ | $3.072 \times 10^{-9}$ |

On the other hand, when 30 MeV and 60 MeV protons impact the silicon material, inelastic processes dominate, and the elastic collision cross-section becomes very small as the energy of the incident proton increases. Both elastic and inelastic collisions of incident protons lead to the displacement of silicon atoms and the formation of displacement damage defects. The data in Table 4 show that the scattering cross-section of gamma rays is $5.522 \times 10^{-3}$ mb, while the elastic and non-elastic scattering cross-section of 60 MeV protons is $5.778 \times 10^2$ mb, which is $1.05 \times 10^5$ times larger than the scattering cross-section of gamma rays.

The interaction process between protons and gamma rays with the silicon nucleus is described above. Secondary electron production by the Compton effect of gamma rays in silicon, as well as the ionization and excitation process of electrons in the atomic shell caused by protons in the presence of Coulomb gravity, are not considered. This is because displacement radiation damage is primarily caused by atoms leaving their lattice positions, while the loss of outer-shell electrons only results in the formation of electron–hole pairs and does not cause displacement damage, also known as Frenkel defects.

## 5. Conclusions

Through the above experimental procedure, we investigated the proton and gamma radiation effects on reach-though SPADs. We tested and analyzed the changes in SPAD parameters, such as I–V characteristics, gain and spectral response, before and after irradiation under different bias conditions and proton energies. Based on the analysis and summary of the experimental results, we draw the following conclusions:

1.  The leakage current of the SPAD increases with the increase in TID. Under the "ON$_{M=100}$"-bias condition, the increase in the leakage current is significantly larger than that under the "OFF"-bias condition. After 30 krad(Si) gamma irradiation, the leakage current increases to 20.94 nA, resulting in device failure. The change in the leakage current after proton irradiation follows a similar trend as that after gamma irradiation. However, by comparing the parameter degradation laws of protons and gamma rays under the same TID, it is found that the leakage current of the device is more sensitive to the displacement radiation damage caused by protons.

2.  Under "OFF"-bias conditions, the gain after gamma irradiation is almost unchanged, meaning that there is no significant change in photocurrent generation. However, after proton irradiation, both the multiplied dark current and the non-multiplied dark current increase significantly compared to before irradiation, leading to an increase in gain.

3.  The spectral response after gamma irradiation exhibits fluctuations in the wavelength range of 700–900 nm, while it remains unchanged in other wavelength ranges. When

the proton irradiation fluence reaches $1 \times 10^{11}$ p/cm$^2$, the displacement damage defects affect the photoelectron multiplication process, resulting in a decrease in the photocurrent and a decrease in the spectral response.

4. The cross-section for the formation of displacement damage defects by protons in silicon is $1.05 \times 10^5$ times greater than the scattering cross-section of gamma rays. The elastic collisions and nuclear reactions between protons and silicon atoms are the main causes of displacement damage in silicon. For SPADs, their parameter degradation is more sensitive to displacement radiation damage.

5. Gamma and proton tests conducted under GTO altitude, a four-year on-orbit mission, and two shielding conditions indicate that the SPAD device will not experience significant performance degradation in leakage current, gain and spectral response parameters. Further testing is required to verify other parameters such as dark count rate (DCR) and photon detection probability (PDP) in these conditions.

**Author Contributions:** Conceptualization, M.X. and Y.L.; methodology, M.X. and Y.L.; software, M.X.; validation, M.X. and Y.L.; formal analysis, Y.L.; investigation, M.X. and Y.L.; resources, M.X. and M.L.; data curation, M.X.; writing—original draft preparation, M.X.; writing—review and editing, M.X. and Y.L.; visualization, M.X.; supervision, Y.L.; project administration, Y.L.; funding acquisition, Y.L. All authors have read and agreed to the published version of the manuscript.

**Funding:** This research was funded by The Chinese Academy of Sciences "Western Young Scholars" Program under Grant No. 2021-XBQNXZ-020, Xinjiang "Tianshan Talents" training program of top young talents project under Grant No. 2023TSYCCX0106.

**Data Availability Statement:** Data are contained within this article.

**Conflicts of Interest:** The authors declare no conflicts of interest.

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
