# Peer review of "Comparison of Proton and Gamma Irradiation on Single-Photon Avalanche Diodes"

_electronics, doi:10.3390/electronics13061086_

Round 1

Reviewer 1 Report

Comments and Suggestions for Authors

In their paper, the Authors report the results of their study on the effects of proton and Gamma rays on SPADs. They find more serious effects in the case of photon irradiation, but, under conditions corresponding to realistic orbital operation, an overall small degradation of the device performance.

Even though the content of the paper is interesting and worth of publication, I feel that some important points of the text should be clarified.

In particular, the Authors seem to distinguish between a the "bias voltage" applied to the device and a "bias condition" used to modify the gain M. For example, Figures 4 and 5 contain results for two different "bias voltages", but then each of them reports the bahavior of the current as a function of the "reverse bias"; or on line 176 I read "the leakage current at a reverse bias voltage of 200 V under zero bias conditions"; and so on. There seem to be two different biases; however, in a diode the applied voltage should be a single one; do the two "biases" differ for the involved time scale or for what? I don't understand. Probably I lack some information, but the text does not help the reader to understand this point.

Also the English form does not help: in several sentences of the text the verb is missing or it is used in a wrong form.

Therefore, I suggest major revisions for this manscript, and in the following I detail the most critical points.

1) First of all, as I have written, the Authors should clarify the distrinction they do between these two "biases" and explain exactly how they change the gain M of the device. In particular, they should improve the sentences on the lines 18-19, on the lines 149-151, on the lines 176-177 and, more in general, of the overall "Results" section. Also a definition of the gain M would be useful: is it the definition of line 264 in the overall paper or not?

2) lines 15-16: the meaning of "TID" and "GTO" should be clarified here (the first time they are used, rather than in the text)

3) lines 18-19: also the English form should be corrected: does "is more serious when the working bias with the gain of 100" mean "is more serious for the working bias corresponding to the gain of 100"?

4) line 23: replace "are" with "being"

5) line 25: replace "SPADs device" with "SPAD devices"

6) line 32: replace "incident" with "impacts"

7) line 33: insert "experience a" before "continuous"

8) line 34: remove "an" before "active"

9) line 68: replace "causing" with "cause"

10) line 81: remove "a" before "back-illuminated"

11) line 88: replace "cross-sectional" with "cross-section"

12) line 89: replace "with" with "have"

13) line 95: "low-doped" or "high-doped" (p+)?

14) line 97: replace "incident" with "impact" (or "impinge")

15) line 97: replace "generates" with "generate"

16) line 101: inset something like "moving them" before "outside"

17) line 102: replace "multiplied" with "multiply"

18) line 105: replace "absorb" with "absorbing"

19) line 109: replace "remained defects in" with "remaining defects in the"

20) line 111: insert "of" after "increase"

21) line 132: remove the comma after "DDD"

22) line 145: explain "NIEL" (non-ionizing energy loss)

23) correct the sentence on the lines 171-173: for example: "the leakage current, which is measured in the dark environment, increases slightly with the increase of radiation dose"

24) line 202: insert "which" before "means"

25) lines 266..: I don't understand why the term "unit" is used to indicate the values without bias voltage (this is just a curiosity)

26) lines 317-318: maybe replace "process , detailed calculations with the" with "process; through detailed calculations, the"

27) line 341: replace "incident" with "impinges"

28) correct the sentence on the lines 343-345 (maybe replace ", as presented" with "we obtain the data reported"?)

29) line 351: replace "incident" with "impinge"

30) line 354: replace "forming" with "form"

31) line 380: insert "which" before "means"

Comments on the Quality of English Language

In several sentences of the text the verb is missing or it is used in a wrong form. I have specified them in the previous field.

Author Response

This paper has been revised and modified for English grammar, and the English writing has been improved, with smoother and clearer expressions.

1) First of all, as I have written, the Authors should clarify the distrinction they do between these two "biases" and explain exactly how they change the gain M of the device. In particular, they should improve the sentences on the lines 18-19, on the lines 149-151, on the lines 176-177 and, more in general, of the overall "Results" section. Also a definition of the gain M would be useful: is it the definition of line 264 in the overall paper or not?

The radiation bias condition seems not described clearly,we modified it and use “ON”-bias and “OFF”-bias conditions to describe the two biases and the definition of “ON”-bias and “OFF”-bias is identified clearly in lines 152-156 the sentence of lines 18-19, lines 149-151,  lines 176-177 and "Results" section are also modified.

2) lines 15-16: the meaning of "TID" and "GTO" should be clarified here (the first time they are used, rather than in the text)
modified

3) lines 18-19: also the English form should be corrected: does "is more serious when the working bias with the gain of 100" mean "is more serious for the working bias corresponding to the gain of 100"?

The sentence has changed to "Compared with the “OFF”-bias conditions, the leakage current degradation is more serious than “ON”-bias and its more sensitive to displacement radiation damage caused by protons com-pared with gamma rays under the same TID. "

4) line 23: replace "are" with "being"
replaced

5) line 25: replace "SPADs device" with "SPAD devices"
replaced

6) line 32: replace "incident" with "impacts"
replaced

7) line 33: insert "experience a" before "continuous"
replaced

8) line 34: remove "an" before "active"
removed

9) line 68: replace "causing" with "cause"
replaced

10) line 81: remove "a" before "back-illuminated"
removed

11) line 88: replace "cross-sectional" with "cross-section"
replaced

12) line 89: replace "with" with "have"
replaced

13) line 95: "low-doped" or "high-doped" (p+)?
low-doped(P),the doping concentration is about 1E16cm-3

14) line 97: replace "incident" with "impact" (or "impinge")
replaced

15) line 97: replace "generates" with "generate"
replaced

16) line 101: inset something like "moving them" before "outside"

The sentence has been modified to "The strong electric field (E>1×105V/cm) in the multiplication region accelerates the photoelectrons to obtain sufficient kinetic energy, and the photoelectrons transfer energy to the electrons of new lattice atoms, ionizing them into secondary electrons by the Coulomb interaction."

17) line 102: replace "multiplied" with "multiply"
replaced

18) line 105: replace "absorb" with "absorbing"
replaced

19) line 109: replace "remained defects in" with "remaining defects in the"
replaced

20) line 111: insert "of" after "increase"
inserted

21) line 132: remove the comma after "DDD"
removed

22) line 145: explain "NIEL" (non-ionizing energy loss)
modified

23) correct the sentence on the lines 171-173: for example: "the leakage current, which is measured in the dark environment, increases slightly with the increase of radiation dose"

the sentence is modified to "Figure 4 (a) shows that the leakage current increases with increasing radiation dose under the “OFF”-bias condition. This is measured in a dark environment. "

24) line 202: insert "which" before "means"
modified

25) lines 266..: I don't understand why the term "unit" is used to indicate the values without bias voltage (this is just a curiosity)

This is a general formula, UNIT represents a unit gain of 1. In this paper, a gain of 1 is used when the bias voltage is 0 V

26) lines 317-318: maybe replace "process , detailed calculations with the" with "process; through detailed calculations, the"
modified

27) line 341: replace "incident" with "impinges"
replaced
28) correct the sentence on the lines 343-345 (maybe replace ", as presented" with "we obtain the data reported"?)
replaced
29) line 351: replace "incident" with "impinge"
replaced
30) line 354: replace "forming" with "form"
replaced
31) line 380: insert "which" before "means"
inserted

Reviewer 2 Report

Comments and Suggestions for Authors

This paper examines the performance of single-photon avalanche diodes (SPADs) following exposure to proton and gamma irradiation. It concludes that, under the specified shielding conditions, SPAD devices are likely to maintain their performance without significant degradation while in orbit. Below are my comments:

1.  Please provide the full names for SRH, NIEL, and any other abbreviations that are not spelled out in the text.

2.   In Figure 8, the responsivity appears to fluctuate within a certain range rather than showing an actual decrease following gamma/proton irradiation. Have you conducted repeated measurements and included the standard deviation to support this observation?

3.     Line 21-23: Further analysis showed that the non-elastic and elastic cross section of proton incident silicon, which caused displacement damage defects, is 10000 times that of gamma rays,

Line 388-390: The reaction cross-section for the formation of displacement damage defects formed by elastic collisions between gamma rays and silicon nuclei is 100000 times lower than that of protons.

This presents a discrepancy between the two sections. Could you please verify the correct factor—whether it's 10,000 or 100,000—and provide further clarification on this matter?

4.   Could you please compare the afterpulsing behavior of SPADs before and after radiation exposure? This comparison would serve as a valuable method to assess their performance.

Author Response

This paper has been revised and modified for English grammar, and the English writing has been improved, with smoother and clearer expressions.

Comments and Suggestions for Authors
This paper examines the performance of single-photon avalanche diodes (SPADs) following exposure to proton and gamma irradiation. It concludes that, under the specified shielding conditions, SPAD devices are likely to maintain their performance without significant degradation while in orbit. Below are my comments:

1.  Please provide the full names for SRH, NIEL, and any other abbreviations that are not spelled out in the text.

The full names of SRH,NIEL,LIDAR,PSTAR,CERN has been added.

2.   In Figure 8, the responsivity appears to fluctuate within a certain range rather than showing an actual decrease following gamma/proton irradiation. Have you conducted repeated measurements and included the standard deviation to support this observation?

The gamma irradiation in Fig. 8a shows a fluctuation in the spectral response in the range of 700 nm to 900 nm, but the proton shows a decrease in the spectral response after irradiation at a fluence of 1E11p/cm-2. We have also found a spectral decrease after neutron irradiation of APDs in the published literature "Numerical simulation of neutron radiation effects in avalanche photodiodes", Fig.7. Similarly, the spectral response of a PIN device after gamma irradiation is shown in the paper "Spectral Response of a Gamma and Electron Irradiated Pin Photodiode" Fig.1. This indicates that the spectral response of the APD device after gamma and proton irradiation would be decreased. But the DDD is too high for the spads.
In order to verify the reliability of the measurements, we performed spectral response curves for 60 MeV protons also at 1E10, 1E11 p/cm2 fluxes, which are provided in the article and the previous spectral response curves of 30MeV was replaced by 60 MeV protons. The results similarly showed a obvious decrease in spectral response at fluxes up to 1E11p/cm-2.

3.     Line 21-23: Further analysis showed that the non-elastic and elastic cross section of proton incident silicon, which caused displacement damage defects, is 10000 times that of gamma rays,Line 388-390: The reaction cross-section for the formation of displacement damage defects formed by elastic collisions between gamma rays and silicon nuclei is 100000 times lower than that of protons.This presents a discrepancy between the two sections. Could you please verify the correct factor—whether it's 10,000 or 100,000—and provide further clarification on this matter?

Detailed calculations were verified and errors in Line 21-23 、 Line 397-401 has been corrected.

A sentence has been added in line361-364 “ The data in Table 4 shows that the scattering cross section of gamma rays is 5.522E-03 mb, while the elastic and non-elastic scattering cross section of 60 MeV protons is 5.778E+02 mb, which is 1.05E5 times larger than the scattering cross section of gamma rays.”

4.   Could you please compare the afterpulsing behavior of SPADs before and after radiation exposure? This comparison would serve as a valuable method to assess their performance.

The afterpulsing behavior of SPADs was not measured in this comparative experiment of proton and gamma radiation. But this measurement will be performed in the next article.

Reviewer 3 Report

Comments and Suggestions for Authors

1    In line 110, SRH should give the full name of Shockley–Read–Hall since this is 1st appear in this paper.

2.      In line 114, the author says that SPAD was selected for comparative testing, but I don’t know where this testing is in this paper.

3.      Fig. 1 should label the dimensions of this device.

4.      The explanation between line 133 and 135 said that “TID and DDD corresponding to the to-133 tal proton flux used in the experiment are greater than the shielded design values shown”, how did the author get this result, for example, the reference or just simulation?

5.      In line 149, all pins are shorted. Are these pins grounded or just floating? If just floating, it is not correct for zero bias condition.

6.      Please describe the gamma radiation experiment condition in “Experiment Design

7.      In Fig. 4a, can you give out a zoom in figure to show the region around reverse bias of 200V?

8.      In Fig. 5, you just show the 5E10 in the label, but you mention it is 1E11 p/cm2 (line 202), could you explain this difference? Also, you should check this unit through the paper, there should be a space between the number and unit.

9.      Can you give the bias voltage corresponding to the M =10 and 100(line212)?

10.  In Fig.6, this line style is so band that we cannot distinguish what every line stands for.

11.  In line 259 -260, “photocurrent with and without bias volt- 259 age does not change significantly after irradiation”, why do the authors give this conclusion?

12.  In line 261-262, “The increase in 𝐼𝑢𝑛𝑖𝑡,𝑑𝑎𝑟𝑘 is  even more significant” , why do the authors give this conclusion?

13.   In Fig. 8a, can you give the zoom in inset figure between 700 – 900nm to show the difference clearly? Also, Fig. 8 label is too small.

14.  In line 295-298, this description is somehow meaningless.

15.  In line 333, the authors said that this SPAD device does not feature a large Si-SiO2 interface. Since it does not have si-sio2 interface, why explain lines between 308-310.

16.   Reference format should be checked thoroughly. For example, Ieee should be IEEE with all letters capitalized. Also, the journal name should use abbreviations or full names uniformly, please do not mix these two formats.

Comments on the Quality of English Language

1. English should be improved somehow, for example in Line 265, you use "is refers to" 

Author Response

 This paper has been revised and modified for English grammar, and the English writing has been improved with smoother and clearer expressions.

1    In line 110, SRH should give the full name of Shockley–Read–Hall since this is 1st appear in this paper.
The full names of SRH,NIEL,LIDAR,PSTAR,CERN has been added.

2.      In line 114, the author says that SPAD was selected for comparative testing, but I don’t know where this testing is in this paper.
The sentence description is inaccurate and has been modified to“A reach-through structural SPAD was selected for gamma and proton radiation effect research due to its simple structure, mature manufacturing process, and complete the-oretical model for analyzing I-V and spectrum characteristics.”

3.      Fig. 1 should label the dimensions of this device.
The dimensions of this device has been added in Fig. 1

4.      The explanation between line 133 and 135 said that “TID and DDD corresponding to the total proton flux used in the experiment are greater than the shielded design values shown”, how did the author get this result, for example, the reference or just simulation?

 The calculated TID and DDD of SPAD in orbit is 1.35krad and 2.84E7MeV/g, the TID and DDD we used in the experiment is higher than that in Table 2.To avoid ambiguity, we changed the sentence in Line 130" In addition to particles produced by solar protons, protons and electrons from the Van Allen radiation belts and solar flare outbursts also contribute to TID and DDD. The contribution of radiation belts and solar flare outbursts to TID and DDD was taken into account in the design of this experiment. Therefore, the TID and DDD corresponding to the total proton flux used in the experiment exceed the shielding design values shown in Table 1, as detailed in Table 2."

5.      In line 149, all pins are shorted. Are these pins grounded or just floating? If just floating, it is not correct for zero bias condition.
The pins are shorted and grounded,we modified the sentence in line 153

6.      Please describe the gamma radiation experiment condition in “Experiment Design”
The sentence has been added in line 148 "The gamma irradiation was conducted at the Co-60 gamma source of Xinjiang Institute of Physics and Chemistry, Chinese Academy of Sciences. The dose rate was 50 rad(Si)/s, with a cumulative dose of 70 krad(Si). "

7.      In Fig. 4a, can you give out a zoom in figure to show the region around reverse bias of 200V?
An enlarged figure has been added in Fig. 4a

8.      In Fig. 5, you just show the 5E10 in the label, but you mention it is 1E11 p/cm2 (line 202), could you explain this difference? Also, you should check this unit through the paper, there should be a space between the number and unit.
A mistakes in Lin 204 has been modified, and the I–V characteristics of SPAD after 1E11 p/cm2 proton irradiation is in figure 6. The space has been added between number and unit.

9.      Can you give the bias voltage corresponding to the M =10 and 100(line212)?
The bias voltage has been added in Line 152 to 157

10.  In Fig.6, this line style is so band that we cannot distinguish what every line stands for.
A clear diagram is presented in Fig.6

11.  In line 259 -260, “photocurrent with and without bias voltage does not change significantly after irradiation”, why do the authors give this conclusion?
This conclusion comes from the results of our experimental measurements, by observing the change of photo current and dark current before and after irradiation, we found that both photo current and dark current are increasing after irradiation, but the increase of dark current is more obvious, so it leads to a larger gain.

12.  In line 261-262, “The increase in ?????,???? is  even more significant” , why do the authors give this conclusion?

This conclusion also comes from the results of our experimental measurements, The increase in ?????,???? is  even more significant than ?????.
To avoid ambiguity, we changed the sentence in Line 262-265"By observing the change in photocurrent and dark current before and after irradiation, it was found that both currents increased after irradiation. However, the dark current I_dark and the I_(unit,dark) both increased significantly compared with those before irradiation."

13.   In Fig. 8a, can you give the zoom in inset figure between 700 – 900nm to show the difference clearly? Also, Fig. 8 label is too small.
A clearer diagram was substituted and the label was adjusted

14.  In line 295-298, this description is somehow meaningless.
The sentence was deleted.

15.  In line 333, the authors said that this SPAD device does not feature a large Si-SiO2 interface. Since it does not have si-sio2 interface, why explain lines between 308-310.

The lines between 308-310 was deleted.

16.   Reference format should be checked thoroughly. For example, Ieee should be IEEE with all letters capitalized. Also, the journal name should use abbreviations or full names uniformly, please do not mix these two formats.

The reference format have been modified. 

Round 2

Reviewer 1 Report

Comments and Suggestions for Authors

Now the paper is much clearer and, in my opinion, can be accepted for publication.

In the following I report a few minor corrections that I suggest to do:

a) line 96: replace "generates" with "generate";

b) line 180: replace "increase of by" with "increase by";

c) figure 8: add "(a)" and "(b)" inside the two panels;

d) line 318: replace "transportation, through" with "transportation: through";

e) line 355: replace "shows" with "show".

Author Response

a) line 96: replace "generates" with "generate";
replaced

b) line 180: replace "increase of by" with "increase by";
replaced

c) figure 8: add "(a)" and "(b)" inside the two panels;
added

d) line 318: replace "transportation, through" with "transportation: through";
replaced

e) line 355: replace "shows" with "show".
replaced

Reviewer 3 Report

Comments and Suggestions for Authors

1. Just correct wrong tab of Ref[11] 

Author Response

  1. Just correct wrong tab of Ref[11] 

Wrong tab of Ref[11] has been corrected.
